# Internal Quality Controls in the Medical Laboratory: A Narrative Review of the Basic Principles of an Appropriate Quality Control Plan

**DOI:** 10.3390/diagnostics14192223

**Published:** 2024-10-05

**Authors:** Lorenz Gruber, Artur Hausch, Thomas Mueller

**Affiliations:** 1Department of Laboratory Medicine, Hospital Gmunden, A-4810 Gmunden, Austria; 2Department of Laboratory Medicine, Hospital Voecklabruck, A-4840 Voecklabruck, Austria

**Keywords:** bias (MeSH ID: D015982), diagnostic errors (MeSH ID: D003951), laboratories, clinical (MeSH ID: D000090464), quality control (MeSH ID: D011786), risk assessment (MeSH ID: D018570)

## Abstract

To ensure the quality of their analyses, medical laboratories carry out internal quality control (IQC) on a daily basis. IQC involves control samples with known target values for all parameters used by a laboratory in clinical practice. The use of IQC enables the laboratory to monitor the accuracy and precision of laboratory results. The use of appropriate IQC strategies has been accepted in medical laboratories for decades, and IQC has been included in international recommendations and guidelines. The term “IQC strategy” (also termed a quality control plan) refers to the types of IQC materials to be measured, the frequency of IQC events, the number of concentration levels in each IQC event, and the IQC rules to be used. A scientifically sound IQC strategy must follow two principles, namely, (1) statistical follow-up on the IQC results generated in the laboratory by means of Levey–Jennings control charts and Westgard rules (i.e., quality control by means of statistical procedures) and (2) the determination of limits on the basis of medical considerations and the definition of analytical goals (quality control on the basis of medical relevance). In this narrative review, we describe the fundamental principles of an adequate IQC strategy for laboratorians and nonlaboratorians.

## 1. Introduction

In a medical laboratory, analyses are performed to support medical decisions [1,2]. Laboratory results from patients aid in diagnosis, prognosis assessment and therapy monitoring [1,2,3,4]. Clearly, the laboratory values obtained must be analytically correct [1,2]. The analytical process must therefore focus on adequate patient care [1,2,3,4].

To ensure the quality of their analyses, medical laboratories periodically participate in external quality assessment programs and use internal quality control (IQC) [2,5,6,7]. In clinical practice, every laboratory test must undergo a daily testing program with respect to analytical quality, which is performed with the help of IQC. IQC involves samples (usually obtained from selected suppliers) with known value assignments for all parameters used by a laboratory in clinical analysis. The use of IQC allows the accuracy and precision of the results to be monitored. The use of appropriate IQC strategies has been generally accepted for decades [2,5,6,7,8,9], and IQC has also been included in international recommendations and guidelines [10,11,12,13].

Currently, the term “IQC strategy” (often also referred to as a quality control plan) encompasses the adequate application of IQC for all parameters determined for medical purposes in the respective laboratory. For each laboratory parameter, the IQC strategy must define (1) the types of IQC materials, (2) the frequency and time points of IQC measurements, (3) the number of concentration levels, and (4) the IQC rules to be used (rejection rules and warning rules) in conjunction with the consequences of rule violations (including troubleshooting and a clinician notification plan in case of erroneous patient results) [2,5,8,9,10,11,13]. A well-designed IQC strategy should reliably detect a change in the performance of a laboratory test that may cause a risk of (unintended) harm to a patient, based on the intended medical use of the results. The laboratory director is responsible for the implementation of an appropriate IQC strategy [5,10].

Although the application of suitable IQC strategies in medical laboratories is anchored in international recommendations and guidelines, to our knowledge, scientifically sound IQC strategies are currently not implemented or are insufficiently implemented in many laboratories. This paper therefore aims to summarize, through a narrative literature review, what can be gleaned from the scientific literature with respect to the implementation of IQC in a medical laboratory. We describe only the basics here; for more in-depth considerations, please refer to further literature.

## 2. Fundamentals and Overview

In previous decades, medical laboratories mainly practiced so-called batch testing. At that time, the IQC strategy focused on the questions of how many IQC levels should be measured and which IQC rules should be applied [8]. IQC was scheduled at certain points in a batch of patient samples (usually at the beginning and end), and measurements were made for a single run. The IQC results were then used to determine whether the patient results for this run were acceptable [5,8]. This practice is usually called “batch” quality control [14].

However, since the introduction of continuous-production analyzers in medical laboratories, the question of when or how often IQC should be performed has also become important with regard to determining an appropriate IQC strategy [8,9,10]. In contrast to batch testing, the IQC of continuous-production analyzers is only an indication of how well or how poorly the measurement method is performing at the time IQC is performed [8,9,10]. In continuous-production analyzers one distinguishes between “bracketed” quality control and “critical control point” quality control [14].

Another recent trend with respect to the IQC strategy of a medical laboratory is dealing with the principles of risk management in the analytical process. The goal is to avoid unintended patient harm due to erroneous laboratory results [15,16].

A medical laboratory’s IQC strategy should be based on quality standards and specified patient safety requirements [7,10,15,16]. For more than 40 years, the scientific literature has explicitly stated that the monitoring of IQC results should be carried out via Levey–Jennings control charts (see Figure 1) and the so-called Westgard rules (i.e., “quality control using statistical methods”) [5,9,10,17,18]. In a second step, the laboratory must also set limits for IQC results on the basis of medical factors (i.e., “quality control based on medical relevance” with definitions of analytical goals) [5,7,10,19,20,21,22,23,24].

Traditionally, numerous key figures have been used to check the QC results for each individual measurand (referred to as laboratory parameters in this paper for simplicity), namely, the mean, bias, standard deviation (SD) or coefficient of variation (CV), allowable total error (TEa) and sigma metric [2,5,6,7,9,10,19,20,21,22,23,24,25].

At this point, we would like to emphasize that once a QC strategy has been defined, it should not be kept unchanged over a long period; instead, a continuous improvement process is explicitly called for in the scientific literature. To maximize the error detection rate in the analytical process while minimizing false error signals, continuous evaluation of a laboratory’s IQC program is required [5,9,10,24]. Indeed, the laboratory director and staff must ensure that the IQC strategy is fulfilling its intended purpose through constant re-evaluation [9,10,24].

## 3. Mean and Bias

We calculate the (arithmetic) mean as the sum of the values of a variable in a sample divided by the number of values [10]. The mean describes the center of a distribution via a numerical value and thus represents a measure of central tendency. For example, the mean of the IQC results for a certain laboratory parameter at a given level is used as the measure of the central tendency of the concentration level of the IQC material measured over time (i.e., the lab mean) [2].

When a constant mean value is used as a measure of central tendency in IQC, this mean remains the same over time (it is also referred to as a fixed mean). In contrast, when a floating mean value is used, a new (updated) mean value is determined each time IQC is performed and is therefore applied as a constantly changing measure of the central tendency (also referred to as the cumulative mean or the moving average) [2,10].

The difference between the “true” value of an IQC and the lab mean of this IQC is called the bias. The bias thus serves as a measure of accuracy, i.e., it estimates the (average) systematic error of the respective IQC material for a laboratory in a certain period [5,6,10].The problem with IQC samples, however, is that the “true” value for a particular laboratory parameter is often not known. To overcome this problem, some manufacturers of IQC materials have established so-called peer groups. All laboratories in a peer group have the same analytical methods, the same equipment, the same reagents and the same controls (with the same lot number). Periodically, the IQC results from the different laboratories are collected by the respective manufacturer and evaluated using specific IT tools. With this approach, we can also use mean values to determine a measure of the central tendency of the IQC results in peer groups (i.e., the group mean) [2]. If the peer group is large enough and the data quality is adequate, then we can consider the group mean to be a good approximation of the “true” value.

Consequently, the mean value for a given concentration level of a given IQC material for a laboratory parameter in a medical laboratory (i.e., the lab mean) can be compared to the mean IQC value of a peer group (i.e., the group mean) [2]. The difference between these two mean values is a common measure of the magnitude of the deviation of the mean concentration level for a laboratory compared with that of a peer group [2,10]. This deviation of the lab mean from the group mean is also referred to as the bias (i.e., the systematic measurement error) [2,10,23].

The bias (expressed in %) is therefore calculated for IQC peer groups as follows [2,25]:$$\mathrm{bias}=\frac{\mathrm{lab}\;\mathrm{mean}-\mathrm{group}\;\mathrm{mean}}{\mathrm{group}\;\mathrm{mean}}$$

## 4. Standard Deviation and Coefficient of Variation

The SD is a measure of the variance of data, in this case, the IQC results of the respective laboratory parameter measured over time at a certain concentration level. The SD indicates the extent to which the measured values deviate from their mean values. It is therefore used to determine the imprecision of the measurement method [10]. 

The formula for the SD (expressed in the unit of measurement) is as follows:$$\mathrm{SD}=\sqrt{\frac{\sum {\left(x_{n}-\bar{x}\right)}^{2}}{n-1}}$$
where x_n_ represents the measured values of the respective IQC, $\bar{x}$ is the mean of the IQC, and n is the number of measured IQC values.

To display the IQC results over time, the SDs of the respective measurement method (and the double and triple of the SD) are plotted in the aforementioned Levey–Jennings control charts. These ranges are normally referred to as the 1 s range, the 2 s range and the 3 s range. The SD is also used as the basis for some of the so-called Westgard rules (the details of which are given later).

Another way of expressing the variance of data (i.e., the imprecision) for a measurement method is to use the CV [2,10].

The CV (expressed in %) of IQC is calculated via the following formula [2,24]:$$\mathrm{CV}=\frac{\mathrm{SD}}{\mathrm{lab}\;\mathrm{mean}}\;\cdot \;100$$

The SD or the CV serves as a measure of precision for estimating the random error (i.e., imprecision) of the respective IQC measurement over time [5,6,10].

## 5. Analytical Goals and Allowable Total Error

In addition to “quality control by statistical methods” via Levey–Jennings control charts and Westgard rules, medically relevant target ranges must be defined for each laboratory parameter. This procedure is termed “quality control based on medical relevance”. The medically relevant range is referred to as the analytical goal or TEa [2,5,7,10,19,20,21,22,23,24]. An analytical goal corresponds to a medically relevant range that is defined around the “true” value of a specific IQC measurement [2,7]. Consequently, the TEa is used to define the maximum permissible analytical error for the IQC of a specific laboratory parameter that may occur without impacting medical decisions [2,5,6,7,10,19,20,21,22,23,24].

The TEa is therefore a commonly used indicator that defines the quality requirement for the medical use of a laboratory parameter. There are (unfortunately) no generally accepted thresholds for the TEa, but it is the task of the laboratory director to define the maximum analytical error that may occur for each laboratory parameter (and if necessary, for different analyte concentrations) without influencing medical decisions [2,5,6,7,10,19,20,21,22,23,24].

The literature describes different approaches to this task. Currently, the following hierarchical classification, in descending order of preference, is common [10,20,21,22,23,26,27,28]: outcome studies (this is the first model and theoretically the best model; however, there are only studies for a few laboratory parameters); biological variation (this is the second model, in which the analytical error should be smaller than that of biological variation; more detailed information can be obtained from the online “EFLM Biological Variation Database”); and state-of-the-art methods (this is the third model, in which the performance of a measurement procedure is defined as acceptable; this acceptable performance represents the “best” level that can be achieved with current technology and/or a level that is similar to the performance of peer groups).

Some approaches are more appropriate for certain laboratory parameters than others are. Regardless of which model is used, the laboratory director should consult with clinicians to determine an appropriate TEa for each laboratory parameter for the respective patient population [10].

Consideration of the above factors yields a “Levey–Jennings Control Chart”, as shown in Figure 2.

## 6. Sigma Metric

A possible way to characterize the stable performance of a measurement system in relation to the medical quality requirement for a laboratory parameter (i.e., the TEa) is to use the sigma metric [7,9,10,24]. In general, a higher sigma indicates a lower error rate of the analytical process, whereas a lower sigma for a given TEa is an indicator that the selection of IQC rules needs to be changed and/or that more frequent IQC measurements are necessary to detect errors in the analytical process [7,10,24]. Although the sigma has several disadvantages [10], knowledge of the sigma of a laboratory parameter is important because a laboratory can use this metric to guide its IQC strategy according to medical needs [7,10,19,24]. The sigma metric is an integrated key figure that includes the bias, CV and TEa [7,10,19,24].

The sigma metric is usually calculated via the following formula [7,9,24,25]:$$\mathrm{Sigma}\;\mathrm{metric}=\frac{\mathrm{TEa}\;\left(\mathrm{expressed}\;\mathrm{in}\;\%\right)-\mathrm{bias}\;\left(\mathrm{expressed}\;\mathrm{in}\;\%\right)}{\mathrm{CV}\;\left(\mathrm{expressed}\;\mathrm{in}\;\%\right)}$$

A high sigma value, for example, 6, indicates a low error rate, whereas a low sigma value, for example, 3, indicates a much higher error rate [24]. In general, higher sigma values indicate that a less stringent IQC strategy can be applied, and lower sigma values indicate that a measurement method may require more frequent IQC to detect process errors [24].

## 7. IQC Rules (So-Called Westgard Rules)

The goal of quality control via statistical methods in a medical laboratory is to detect changes in the stable operation of a measurement procedure that cause a significant increase in the risk of producing erroneous patient results, which, in turn, could negatively influence medical decision making [9,10,16]. This is called an out-of-control condition.

There are two basic classifications for out-of-control conditions: transient and permanent. Transient conditions may affect a single sample or multiple samples over a short period (e.g., inadequate cleaning, electronic noise or clots or debris in the pipette). Owing to the transient nature of these conditions, they may no longer be present at the next scheduled IQC event and therefore will not be detected. Permanent out-of-control conditions remain until they are detected and the cause is corrected (e.g., calibration problems, reagent deterioration, incorrect fluid levels, incorrect temperature control). Permanent out-of-control conditions can, in turn, be divided into two subcategories: conditions that change the constant error (i.e., bias) of the measurement procedure and conditions that change the random error (i.e., imprecision) of the measurement procedure. IQC rules should therefore recognize changes in both bias and imprecision [29].

An IQC strategy involves, among other things, deciding which IQC rule(s) should be used to evaluate the IQC results [9,10]. An IQC rule is a formal decision process that takes the results of one or more IQC measurements and either determines that the measurement procedure is operating in a stable, controlled state (acceptance of the IQC rule) or that the measurement procedure is not operating in a stable, controlled state (rejection of the IQC rule) [9,10]. A variety of statistical IQC rules have been proposed for use in medical laboratories. These IQC rules are also referred to as Westgard rules, after their inventor James O. Westgard [30]. Certain rule violations indicate a random error, and other rule violations indicate a systematic error [9]. Table 1 shows several statistical IQC rules that are commonly used in medical laboratories (not an exhaustive list).

Depending on the IQC strategy, some of these statistical IQC rules can be used as rejection rules or warning rules. Rejection rules and warning rules can vary across different laboratory parameters in a particular laboratory [9]. If a rejection rule is violated, the analytical process must be interrupted, and the cause of the rule violation must be determined and corrected. Violations of warning rules, on the other hand, do not necessarily result in the immediate termination of the use of the respective measuring system; they are primarily intended to detect problems at an early stage and to intervene preventively if necessary.

Individual IQC rules are often combined into an IQC multirule [9,10]. An IQC multirule rejects if any of the individual IQC rules rejects [9,10]. For example, a 1_3s_/2_2s_/4_1s_ multirule evaluates all three individual rules and rejects if even one of the individual rules rejects.

Of course, the IQC rules are not perfect, because even when they are used in combination, they do not detect all analytical errors in a laboratory (i.e., they produce false negatives), and because they also indicate errors in certain situations even when there is no analytical error (i.e., they produce false positives) [5,9,10,31]. The frequency of false negatives and false positives can also be quantified via the “probability of (error) detection” (P_ed_) and the “probability of false rejection” (P_fr_) [5,9,10,31]. In an ideal world with perfect IQC rules, P_ed_ would be 1 (100%), and P_fr_ would be zero.

For example, a characteristic of 1_2s_ is its relatively high P_fr_ (4.6% when measuring one IQC level, 8.9% when measuring two IQC levels and 13.0% when measuring three IQC levels) [31]. A medical laboratory that uses two IQC levels for 20 different laboratory parameters each and applies 1_2s_ can therefore assume that an out-of-control condition is present for at least one laboratory parameter in each IQC cycle [30]. In our opinion, this unacceptably high P_fr_ is the reason why 1_2s_ should not be used as a rejection rule by default.

## 8. Actions in Out-of-Control Conditions

What should laboratory staff do if IQC has detected an error (possible out-of-control condition)? Simply repeating the IQC test on the vial that was already used is not suitable for determining whether there is a random error, as there is a high probability that the result of the same IQC test will be within the control limits on the second run if it is performed a second time [5,9,24]. A documented, systematic procedure must be used to investigate the error and take appropriate action [5,24]. Otherwise, the entire IQC process is a waste of time and effort [5,9].

Therefore, if an IQC rule evaluation indicates that the measurement process is out of control, the first action should be to measure a fresh vial of the previously used IQC material [5,10]. This will rule out problems that could be caused by compromised IQC materials (e.g., evaporation or improper storage conditions) [5,10,24]. If the out-of-control condition is reproduced with fresh IQC material, this should be treated as a genuine error reflecting an analytical measurement problem [5,10]. If fresh IQC material does not confirm the out-of-control condition and the values thus measured are at the previous level, then the measurement system is acceptable, but the content of the previously used IQC vial was defective.

If an out-of-control condition is actually detected, the first step is to limit the error condition by immediately interrupting patient measurements and/or the output of clinical reports [10,16,24]. Examples of common corrective actions for out-of-control conditions are calibration, the replacement of reagent packs or the replacement of electrodes. Less common occurrences may require an in-depth (technical) investigation to determine the root cause of the failure. Once the root cause has been identified and corrected, IQC should be performed again to ensure that the measurement procedure in question is working properly [10].

In addition, the laboratory must identify and revise incorrectly reported patient results [7,10,16,23]. The medically relevant range of a measurement error that requires a corrected result should be defined for each laboratory parameter (e.g., using the TEa).

The recommended approach is to repeat the testing of patient samples via a properly functioning measurement procedure and to compare these results with the originally reported results [10,24]. Differences between results that exceed the specified medically significant level of change must be corrected in the clinical reports [10,24]. Repeat testing should begin from the time at which the IQC rule rejected and should continue retrospectively until the time at which the out-of-control condition occurred [10,24]. This procedure of repeated measurements is also known as retrospectivity. In retrospectivity, it is common to retest patient samples in batches of 10 [10,24]. If one of the 10 patient results needs to be corrected, the laboratory examines another batch of 10 samples. Retrospectivity continues in batches of 10 until an entire batch is found that no longer requires correction of the results [10,24]. This point in time corresponds approximately to the point in time at which the out-of-control condition occurred [10].

If no patient samples are available for repeat testing, or if the laboratory parameters are unstable and therefore cannot be retested, the laboratory should issue a corrected clinical report stating that the result is invalid [10].

## 9. IQC in Particular Situations

As already emphasized, the medical laboratory must determine the mean and SD of the IQC values by repeated measurements of the IQC materials when the measurement procedure is run under stable conditions [5,10]. If the IQC is accompanied by a product insert with values specified by the manufacturer, these values may be used only as a guide and not as a substitute for the “means” and SDs determined by the laboratory [10].

If no previous data are available, initial estimates of the mean and SD are obtained by measuring at least 20 data points on different days [5,9,10]. The measured values obtained in this first phase should represent the analytical procedure in its stable state [5,9,10]. During the initial phase of routine operation with an initial estimate of the mean and SD, the laboratory should closely monitor its IQC data as they accumulate over time, thus incorporating additional components of longer-term sources of variability into the IQC data [5,10].

If a new batch of reagent or a new batch of existing IQC material is used, there can be a shift in the IQC data. This implies that the mean of the measured IQC values can change significantly, but the SD of the measured IQC values usually remains the same as before (the measurement method with its variability remains the same) [10]. In most cases, 10 measurements on different days are sufficient to determine a mean that can be used as a first approximation [10]. A minimum of 10 days allows relevant sources of daily variability in the analysis procedure to be adequately accounted for [10]. However, there are situations where the laboratory needs to establish a mean more quickly. In such cases, the mean of a few days of measurements can be used [10]. A value obtained in this way must be considered transient and should be updated over time with subsequent IQC measurements in the sense of a floating mean.

The criteria for acceptable IQC results are primarily based on the performance that a measurement procedure can achieve, since the purpose of IQC measurement is to verify that the measurement procedure in question meets its expected analytical performance level [9,10]. However, less stringent IQC acceptance criteria can also be used if the risk of unintended patient harm is maintained at an acceptable level [10]. Importantly, setting stricter Westgard rules and restricting the analytical goals do not change the performance of a measurement procedure. While stricter acceptance criteria allow smaller performance deviations to be detected, they also result in a higher rate of rule violations, which, in turn, leads to increased troubleshooting efforts, repeated measurements and likely delays in reporting patient results. The additional effort involved in tracking these (supposed) IQC errors reduces the efficiency of the measurement method and increases operating costs. Therefore, if the TEa remains constant, the sigma metric for the measurement method also remains constant, regardless of which Westgard rules are applied.

## 10. IQC Material

The control material should have properties that make it possible for the material to provide information about the performance of a measurement method when measuring patient samples. Ideally, the matrix of the IQC material (e.g., serum, plasma or urine) should be the same as that of the patient samples to be measured [5,10,24]. Thus, the IQC material should react the same way as patient samples. In other words, the IQC material should be commutable [29].

A medical laboratory should procure sufficient homogeneous and stable control material that will last for a long period (e.g., one year or longer) if possible [5,10,24]. Using the same batch of IQC material optimizes the ability to establish expected results and evaluation criteria [10]. The longer the same batch of IQC material is used, the less frequently statistical properties need to be established for new batches of IQC material.

The control materials should be different from the calibrator materials to ensure that the IQC results provide an independent assessment of the performance of the measurement procedure as a whole, including the calibration procedure [10,24].

There are different types of control materials for medical laboratories [10,17,24]: (1) control materials produced and supplied by the manufacturer of the measurement procedure itself (which are suboptimal, especially if they mimic the calibrator and may therefore be unable to detect some systematic errors); (2) control materials produced by a producer for the manufacturer of the measurement procedure (such control materials are usually produced on behalf of an instrument or reagent manufacturer); and (3) control materials produced by a third party unrelated to the manufacturer of the measurement procedure or the calibrator used for the measurement procedure (so-called “third party controls”, which are usually preferred because they allow an independent assessment of the performance of the measurement procedure).

For a given laboratory parameter, the appropriate concentrations at which the performance of a method should be monitored are based on both clinical decision limits and the analytical measurement interval [9,24]. Ideally, at least one of the IQC materials used should have a concentration at or near the clinical decision limit of the respective laboratory parameter [9,23]. For most laboratory parameters, a minimum of two concentrations of IQC materials is generally recommended [9].

Control materials are generally stabilized to ensure a long shelf life [9,10,24]. There are two common methods for stabilizing IQC materials: freeze-drying (also known as lyophilization or sublimation drying) and the freezing of liquid materials. Lyophilized materials are the most stable form in which IQC material is supplied [10,24], and they have a long shelf life. Liquid materials are convenient but usually need to be stored frozen [10,24]. Since no reconstitution is needed, the IQC results for liquid materials can be used to determine the inaccuracy of a measurement method very well [10].

## 11. Special Case of Multiple Instruments

Most of the laboratory medicine literature discussing the principles and practices of quality control via statistical methods addresses the case of monitoring the performance of a single measurement procedure with a single instrument [32]. However, many laboratories have multiple instruments of the same type that use the same measurement procedures (i.e., the same assays are used on identical instruments).

The problem of developing appropriate IQC strategies for multiple random-access analyzers measuring the same laboratory parameters with the same assays is an important challenge in the modern laboratory [7,9,10,24,32]. Two identical instruments usually have different analytical performance characteristics. In other words, when the same IQC material is used, different means and different SDs can be observed for the two instruments [9,24,32]. This raises the question of whether it makes sense to use the same IQC mean and the same SD for all devices measuring the same laboratory parameter [9,24,32]. However, as very few studies address the design of IQC strategies with multiple instruments, there are currently no consistent recommendations for this situation [32].

A recently published paper, for example, describes the successful real-world application of an IQC strategy for multiple instruments, measuring the same laboratory parameters and using externally provided quality control targets [33]. The authors of this work applied single Levey–Jennings control charts for multiple instruments using one peer group target mean and uniform control limits for each laboratory parameter [33]. With this approach, the error rate of the IQC strategy was consistent with the authors’ expectations, and unnecessary recalibrations were even reduced without the need to amend any results [33].

In our opinion, the use of a single Levey-Jennings control chart for the IQC results of multiple random-access analyzers might also be advantageous for determining the appropriate measurement uncertainty for a particular laboratory parameter. We expect further studies on this very topical issue in the near future.

## 12. IQC Strategy and Risk Management

As previously emphasized, one goal of a medical laboratory’s IQC strategy must be to reduce the risk of patient harm caused by errors in the analytical process [2,10]. Therefore, the IQC strategy of a medical laboratory must also be based on the principles of risk management [2,8,10,15,16]. The risk of patient harm increases when the IQC strategy cannot (adequately) detect erroneous laboratory results that may have medical consequences [10].

A suitable approach to solving this problem is described in [15,34]. Risk is defined in these studies as a combination of the probability of patient harm and the severity of patient harm [2,8,15,16,34]. A risk assessment is carried out via a five-point scale for both the probability of the occurrence of patient harm and the severity of this patient harm [2,15,34]. In accordance with CLSI EP23-A, a corresponding matrix (see Table 2) is then used to determine whether the risk for the medical laboratory is classified as acceptable or unacceptable [15].

In more recent literature, the term “Risk Management Index” (RMI) has also been introduced in this context [2,35,36]. The RMI is the predicted probability of harm divided by the acceptable probability of harm [2,35,36].

Recent publications indicate that an RMI ≤ 1 should be the goal [2,35,36]. An RMI ≤ 1 would mean that the capability and reliability of the respective measurement system combined with the IQC strategy of the laboratory keeps the risk of unintended patient harm at an acceptable level [2,35]. In contrast, an RMI > 1 would indicate that the medical laboratory has not reduced the risk of unintended patient harm to an acceptable level [2,35]. For example, if a laboratory parameter has an RMI of 0.72, then the risk for this measurement method is 28% below the target risk [36]. However, if the RMI for another laboratory parameter is 1.56, this results in a risk of 56% above the target risk level [36]. Theoretically, the RMI would therefore make it relatively easy to determine the risk of unintended patient harm caused by the analytical process for many laboratory parameters and to compare them with each other.

## 13. Patient-Based IQC Procedures

Another approach to monitoring the quality of analytical runs is to use patient data [5,24]. This technique often involves the calculation of a moving patient sample average [5,24]. The potential advantages of this approach are that the matrix effects of synthetic control materials can be avoided and that the cost of IQC can be reduced [5,22]. The concept of averaging patient data for IQC purposes is referred to as the “average of normals” (AON) or “average of patients” or “moving average” [5,9,24,37]. In the more recent literature, this system is also referred to as “patient-based real-time quality control” (PBRTQC) [38,39,40].

## 14. National Regulations for IQC

In the United States of America, the nonprofit Clinical and Laboratory Standards Institute (CLSI) has developed recommendations that are intended for medical laboratories but are also recognized by accreditation bodies and United States government agencies [41]. CLSI document C24, “*Statistical quality control for quantitative measurement procedures: principles and definitions*”, describes IQC strategies on the basis of the quality required for the test, the precision and bias observed for the measurement procedure, and the risk of patient harm [10,41]. In addition, CLSI document EP23, “*Laboratory quality control based on risk management*”, states that a medical laboratory should manage risk by implementing an IQC strategy that ensures that the quality of test results is suitable for clinical use [15]. In principle, these two documents contain the evidence-based information that is described in the previous sections.

For the Federal Republic of Germany, the “*Richtlinie der Bundesärztekammer zur Qualitätssicherung laboratoriumsmedizinischer Untersuchungen*—Rili-BÄK” regulates the basic requirements for quality assurance of laboratory medical examinations [11,12]. The current version of the Rili-BÄK is enforced in accordance with the resolution of the Executive Board of the German Medical Association. With respect to the implementation of IQC, the Rili-BÄK prescribes specific instructions for action. For example, a table with error limits (permissible relative deviations) for each laboratory parameter is available for evaluating the results of IQC for quantitative procedures. At the end of a calendar month, the Rili-BÄK stipulates that the relative quadratic mean value of the measurement deviation for a control sample must be calculated [11]. If this value exceeds the specified measurement deviation, the examination procedure must be blocked, and troubleshooting should begin. However, there is no specific reference to the scientific literature published to date in the entire Rili-BÄK [11].

Switzerland stipulates that IQC must be carried out regularly for all medical laboratory analyses in accordance with the specifications of the Swiss Association for Quality Development in Medical Laboratories (QUALAB) as soon as these are billed according to the federal list of analyses or as part of a flat rate per case in accordance with the Federal Health Insurance Act [13]. In addition to the type and frequency of IQC, the QUALAB also regulates the evaluation of the results. The focus here is on the determination of so-called warning and tolerance limits. By default, the corresponding assigned values from the supplier’s package insert are used, but the QUALAB guidelines also recommend that laboratories determine their own mean values and tolerance limits, provided that these values are within the manufacturer’s ranges. A distinction is made between “warning” and “IQC out of control” [13]. In the case of warnings, patient results may be validated, but the subsequent control results are subject to special rules. As soon as a rule violation is classified as “IQC out of control”, measurements may not be released [13].

In Austria, there are no normative specifications from legislative or medical bodies. Although quality assurance is mandatory in medical laboratories according to official regulations, there is no detailed specification regarding a suitable IQC strategy. This is why there are many different approaches to IQC strategies in Austrian laboratories, some of which do not correspond to an evidence-based approach or do so only to a limited extent.

## 15. Specifications from ISO 15189 Standard on IQC

ISO standard 15189 includes requirements for medical laboratories for the planning and implementation of actions to address risks and opportunities for improvement; thus, it specifies requirements for quality and competence in laboratories in many different respects [42]. ISO 15189 is an accreditation standard. Instead of focusing the activities of a laboratory based entirely on the question, “Does a particular activity meet the requirements of ISO 15189?”, David Burnett (who played a leading role in the creation of ISO 15189) describes, in his classic commentary on ISO 15189 entitled “*A Practical Guide to ISO 15189 in Laboratory Medicine*” [43], that the primary aim when applying the standard is to examine the meaning behind the relevant requirement for each standard criterion and to apply it to one’s own laboratory situation in light of the continuous improvement process [44].

ISO 15189 specifies that IQC is an internal procedure of a medical laboratory that monitors the testing process to verify that the system is working correctly and ensures that the results are sufficiently reliable to be released for patient care [42]. There are requirements for the selection of control samples for internal quality assurance (the consideration of clinical decision limits, independence of the control material), for frequency of use, for consequences in the event of a violation of tolerance limits and for regular retrospective evaluation [42]. In its (brief) specifications for IQC, ISO 15189 fully complies with the principles of a science-based IQC strategy, as described in the previous sections.

## 16. Limitations of Current IQC Strategies

In the previous sections, we have tried to present the current state of scientific knowledge with regard to an adequate IQC strategy. For more than 25 years, however, the “classical” system presented in this review has also been criticized for being too complicated and too expensive [45]. Both points of criticism could be the reason why many medical laboratories do not or only partially implement a scientifically sound IQC strategy. Various surveys of laboratory professionals have shown that there is a considerable difference between the scientific discussion of the IQC strategy and the clinical reality in medical laboratories [46,47]. One could say Westgard rules and analytical goals are frequently quoted but comparatively rarely applied in routine medical laboratories.

It is argued that up to 35% of all determinations are IQC measurements and that only less than 1% of all runs are rejected [45]. This, of course, incurs certain costs. Some authors therefore recommend keeping the IQC strategy simple, so that the whole staff can rigorously and easily follow the established protocols and practices [45]. However, the appropriate instruction and adequate training of laboratory staff should be a standard procedure anyway.

An important assumption of the “classical” IQC strategy is that the control samples used will react the same way as patient samples (i.e., the IQC material is commutable) [29]. However, due to matrix alterations of the ICQ material by the manufacturer, the noncommutability of IQC material is a problem [29,48]. A lack of commutability may create a situation where IQC indicates an unacceptable bias that would prompt the rejection of patient results when, in fact, the patient results are not affected, or where IQC will not demonstrate any bias, but the patient results are significantly biased [48]. These commutability issues might be a cause of false alarms and missed errors [29,48].

Taken together, a “classical” IQC strategy certainly has advantages (e.g., it involves well-established practice with standard guidelines and recommendations, most instruments and middleware routinely support this function, and its performance on error detection is independent of the underlying patient population) but also disadvantages (e.g., the high cost of implementation, the intermittent assessment of assay performance, the potential issue with commutability and the lack of correlation with clinical impact) [29]. Thus, some experts in the field call for a continuous monitoring model without the problem of noncommutability [29,37,38,39,40,48]. They argue that a laboratory needs to use a parameter determined from patient results [29,37,38,39,40,48]. This is the basis of PBRTQC [29,37,38,39,40,48]. The proposed advantages of PBRTQC include the potentially continuous assessment of assay performance, the relatively low cost of implementation, and the evaluation of assay performance based on actual patient data without any issues of noncommutability [29,37,38,39,40,48]. However, PBRTQC requires a more in-depth understanding of statistics, and of the biological and analytical characteristics of individual tests, for its implementation and interpretation [29,37,38,39,40,48]. In addition, PBRTQC requires more advanced laboratory informatics to extract data in order to perform analyses in an automated fashion [29]. Thus, in our opinion, we should consider PBRTQC a useful, complementary tool in addition to the “classical” IQC strategy.

## 17. Summary of Recommendations

Our narrative review shows that a scientifically sound IQC strategy must follow two principles, namely, monitoring the IQC results by means of Levey–Jennings control charts and Westgard rules (i.e., quality control by means of statistical procedures) and the definition of analytical goals on the basis of medical factors (quality control on the basis of medical relevance) [5,7,10,19,20,21,22,23,24].

The aim of such an IQC strategy must be to keep the analytical process in the laboratory under control and to detect “out-of-control conditions” with a high degree of accuracy. The IQC rules are not perfect, because even when they are used in combination, they do not detect all analytical errors in a laboratory (they produce false negatives), and because they also indicate errors in certain situations when there is no analytical error (they produce false positives) [5,9,10,31].

Once an IQC strategy has been defined, it should not remain unchanged over a long period; instead, a continuous improvement process should be applied. Continuous re-evaluation of a laboratory’s IQC program is necessary to maximize error detection in the analytical process while minimizing false error signals [5,9,10,24].

The purpose of an IQC strategy is, of course, to minimize the number of incorrect results in clinical work [2,5,6,9,10,15,16]. Incorrectly reported laboratory results can lead to patient harm [2,7,15,16]. The patient could receive the wrong treatment or incorrectly receive no treatment. On the basis of such mistaken laboratory results, the patient could suffer harm [7,15,16].

When an out-of-control condition is detected, the first step is to prevent further error by immediately ceasing patient measurements and/or the output of results [10,16]. Once the error in the measurement system has been corrected, the laboratory must identify and subsequently eliminate erroneous patient results retrospectively [7,10,16,24].

In the continuous improvement process, in addition to considering the question of the “right” IQC rules, one should also consider the types of IQC materials to be measured, the frequency of IQC measurements and the number of concentration levels for each IQC measurement to keep the analytical process in a stable state [5,8,9,10,11,13,24].

Another approach to improving the IQC strategy is to address (clinical) risk management. The risk of patient harm increases when the IQC strategy cannot adequately detect erroneous laboratory results that may have medical consequences [10]. The risk is defined here as the combination of the probability of patient harm and the severity of patient harm [2,8,15,16]. The term “RMI” is discussed in the literature in this context [2,35,36].

Due to some drawbacks of a “classical” IQC strategy using IQC materials obtained from selected suppliers, PBRTQC should be considered as a complementary quality control tool [29,37,38,39,40,48].

All these considerations should be used to continuously improve an IQC strategy once it has been defined in order to optimize patient care. Indeed, the laboratory director and staff must ensure that the IQC strategy is fulfilling its intended purpose through constant re-evaluation [10,24].

## Figures and Tables

**Figure 1 diagnostics-14-02223-f001:**
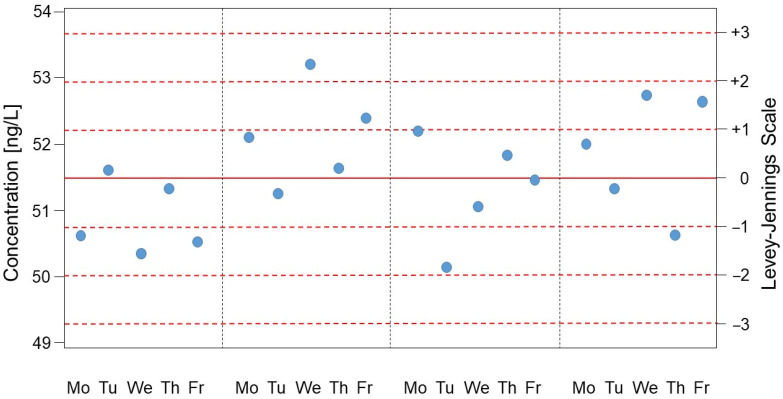
A Levey–Jennings control chart. Figure 1 shows an example of a laboratory parameter over a period of 4 weeks with an IQC on weekdays from Monday to Friday (fictitious data). The abscissa indicates the time course. The left ordinate refers to the concentration of IQC values expressed in ng/L. The right ordinate shows the mean of the IQC values measured in these 4 weeks (i.e., “0” and the corresponding horizontal solid red line) and the standard deviation of the measured IQC values or its multiples (i.e., “−1” to “+1” corresponds to the range of ± one standard deviation, and “−3” to “+3” corresponds to the range of ± three standard deviations; horizontal dotted red lines). The blue dots indicate the measured IQC values.

**Figure 2 diagnostics-14-02223-f002:**
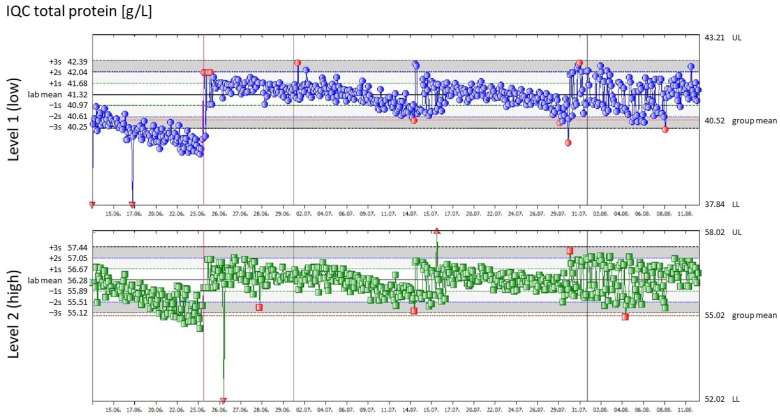
Levey–Jennings control charts for total protein at two concentration levels. Figure 2 shows an example of a Levey–Jennings control chart for total protein over a period of two months at two different IQC levels (actual data from the Voecklabruck laboratory). The abscissa shows the time course. The left ordinate refers to the two mean values of the IQC values measured during these eight weeks, expressed in g/L (“lab mean”), and shows the corresponding 1 s ranges, 2 s ranges and 3 s ranges at the two concentration levels (gray horizontal stripes). The right-hand ordinate shows the target value (mean value of the peer group, i.e., the group mean) with an associated red line and the TEa defined by laboratory management (upper and lower limits, referred to as “UL” and “LL” in the figure). Based on a floating mean, the blue and green dots indicate the IQC values without rule violation, the red dots indicate those IQC values that were rejected by the Westgard rules, and the red triangles indicate those IQC values that were outside the TEa. The gray vertical lines indicate the month break and the red vertical line the time point of a recalculation of the lab mean.

**Table 1 diagnostics-14-02223-t001:** Examples of statistical IQC rules (i.e., Westgard rules).

IQC Rule *	Description of the Rule
1_3s_	One control measurement exceeds $\bar{x}$ ± 3 SD
1_2s_	One control measurement exceeds $\bar{x}$ ± 2 SD
2_2s_	Two consecutive control measurements exceed $\bar{x}$ + 2 SD or $\bar{x}$ − 2 SD
3_1s_	Three consecutive control measurements exceed the same limit, which is either $\bar{x}$ +1 SD or $\bar{x}$ − 1 SD
4_1s_	Four consecutive control measurements exceed the same limit, which is either $\bar{x}$ +1 SD or $\bar{x}$ − 1 SD
R_4s_	The difference between the high and low control measurements within a run exceeds 4 SD
7$\bar{x}$	Seven consecutive control measurements fall on one side of the mean
10$\bar{x}$	Ten consecutive control measurements fall on one side of the mean
7_T_	Seven consecutive control measurements trend in the same direction (i.e., progressively higher or progressively lower)

Abbreviations: IQC, internal quality control; SD, standard deviation; $\bar{x}$ , mean of the IQC. * Some of the statistical IQC rules presented in the medical literature are counting rules. The decision criteria are based on counting the number of IQC results that violate a certain control limit. These types of counting rules can be expressed by abbreviations of the form “A_L_”, where “A” represents the number of control observations and “L” represents a certain control limit [9]. If “A” control observations exceed the limits of the rule, it is determined that the measurement procedure has an out-of-control condition [9].

**Table 2 diagnostics-14-02223-t002:** “Risk acceptability matrix” modified from CLSI EP23-A [13].

Probability of Harm *	Severity of Harm ^†^				
	Negligible	Minor	Serious	Critical	Catastrophic
Frequent	Unacceptable	Unacceptable	Unacceptable	Unacceptable	Unacceptable
Probable	Acceptable	Unacceptable	Unacceptable	Unacceptable	Unacceptable
Occasional	Acceptable	Acceptable	Acceptable	Unacceptable	Unacceptable
Remote	Acceptable	Acceptable	Acceptable	Acceptable	Unacceptable
Improbable	Acceptable	Acceptable	Acceptable	Acceptable	Acceptable

* The five categories of probability of patient harm are assigned as follows (each as “rates over time” and as “proportions of reported patient outcomes”): (1) “frequent” (once a week or ≥1/1000); (2) “probable” (once a month or <1/1000 and ≥1/10,000); (3) “occasional” (once a year or <1/10,000 and ≥1/100,000); (4) “remote” (once every few years or <1/100,000 and ≥1/1,000,000); and (5) “improbable” (once during the entire lifetime of the measurement system or <1/1,000,000). ^†^ The five categories of the severity of patient harm are standardized as follows: (1) “negligible” (inconvenience or temporary discomfort); (2) “minor” (harm or impairment that does not require professional medical treatment); (3) “serious” (harm or impairment that requires medical treatment); (4) “critical” (permanent or life-threatening harm); and (5) “catastrophic” (death of the patient).

## Data Availability

Not applicable.

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
