# Peer review of "Internal Quality Controls in the Medical Laboratory: A Narrative Review of the Basic Principles of an Appropriate Quality Control Plan"

_diagnostics, 2024, doi:10.3390/diagnostics14192223_

Round 1

Reviewer 1 Report

Comments and Suggestions for Authors

This is a useful review of IQC. There are some areas where I believe confusion could be caused.

1. Westgard rules are commonly cited, but not widely used. The commonest rules are, in fact, 2 or 3S. (Rosenbaum et al Quality Control Practices for Chemistry and Immunochemistry in a Cohort of 21 Large Academic Medical Centers; Howanitz et al Clinical laboratory quality control: A costly process now out of control; Westgard The 2017 great global QC survey.)

2. Line 37 - reference sample rather than control

3. The QC material usually does not have known target values. This material is expensive, and the targets, particularly the SD provided by the manufacturer, are wide.

4. Any QC strategy should include patient risk and a documented troubleshooting, patient repeat and clinician notification plan for failed QC.

5. Line 59—usually called bracketed QQC, where a batch of patient samples was released if the QC was in control.

6. line 78. The clinical relevance of an error's impact on patient results is usually a second level of review. The first level is the failure of the statistical rules.

7. Section 3 - Mean and SD. I found this section confusing. The lab mean would be the mean of the instrument or group of instruments in the lab or the network. The terms peer groups and mean refer to EQA subgroups. Controlling individual instruments with their own means and SDs or groups of instruments with a single mean and SD is very topical and should be discussed - (A model for managing quality control for a network of clinical chemistry instruments measuring the same analyte - Giannoli et al.)

8. Line 212 - transient should be random

9. Table 1 - there are extra spaces in the Table between words.

10. There is no need to reproduce the Westgard rules in Figures - they are well known and can be referenced - suggest delete Figures 3,4, 5, 6, 7.

11. Line 314 - repeating the QC sample once, is the most powerful QC rule there is (Should I repeat my 1:2s QC  - Parvin et al)

12. Line 357 - I would not recommend using manufacturers target values. These need to be laboratory defined.

13. Line 394 - noncommutability of QC material is a problem - (Past, present, and future of laboratory quality control: patient- based real-time quality control or when getting more quality at less cost is not wishful thinking Katayev and Fleming).

14. There should be a section on the limitations of QC strategies - cost, false rejections (possible 80% of flags), where QC samples are placed, frequency, not responding to flags, not repeating patient samples - Howanitz et al Clinical laboratory quality control: A costly process now out of control; Westgard The 2017 great global QC survey.

15. Suggest that the Abstrat not be a repeat of sentences from within the text.

Author Response

Comment 1: This is a useful review of IQC. There are some areas where I believe confusion could be caused.

Response 1: We would like to thank Reviewer #1 for this positive statement. We addressed the individual points of criticism in our point-by-point response below.

Comment 2: Westgard rules are commonly cited, but not widely used. The commonest rules are, in fact, 2 or 3S. (Rosenbaum et al Quality Control Practices for Chemistry and Immunochemistry in a Cohort of 21 Large Academic Medical Centers; Howanitz et al Clinical laboratory quality control: A costly process now out of control; Westgard The 2017 great global QC survey.)

Response 2: We appreciate this comment and agree with Reviewer #1, especially for the German-speaking countries. This is one of the main reasons for writing this review. We have included the discussion points raised by reviewer #1 in the newly added section 16 of the revised manuscript.

Comment 3: Line 37 - reference sample rather than control

Response 3: See also the next bullet point. We have rewritten the sentence in line with this point of criticism from Reviewer #1. Firstly, we omitted “control samples” and used “samples” only. Secondly, we have used the term “known value assignments” instead of “known target values” (according to Katayev A et al. J Lab Pec Med 2020;5:28).

Comment 4: The QC material usually does not have known target values. This material is expensive, and the targets, particularly the SD provided by the manufacturer, are wide.

Response 4: See also the previous point. We have rewritten the sentence in line with this point of criticism from reviewer #1. Firstly, we omitted “control samples” and used “samples” only. Secondly, we have used the term “known value assignments” instead of “known target values” (according to Katayev A et al. J Lab Pec Med 2020;5:28).

Comment 5: Any QC strategy should include patient risk and a documented troubleshooting, patient repeat and clinician notification plan for failed QC.

Response 5: We added these considerations to section 1 as suggested by reviewer #1.

Comment 6: Line 59—usually called bracketed QQC, where a batch of patient samples was released if the QC was in control.

Response 6: We considered this point and explained the terms “batch” quality control, “bracketed” quality control and “critical control point” quality control in section 2 of our revised manuscript.

Comment 7: Line 78. The clinical relevance of an error's impact on patient results is usually a second level of review. The first level is the failure of the statistical rules.

Response 7: We have taken this point into account and reworded the respective passage in our revised manuscript. Thus, it should be clear what is the first step and what is the second step.

Comment 8: Section 3 - Mean and SD. I found this section confusing. The lab mean would be the mean of the instrument or group of instruments in the lab or the network. The terms peer groups and mean refer to EQA subgroups. Controlling individual instruments with their own means and SDs or groups of instruments with a single mean and SD is very topical and should be discussed - (A model for managing quality control for a network of clinical chemistry instruments measuring the same analyte - Giannoli et al.)

Response 8: We have understood that we have probably written this passage in a misleading way. In order to make everything clear and unambiguous, we have completely rewritten the relevant paragraph in the revised manuscript (see section 3). For example, in our laboratory we use third party controls from Bio-Rad. When using these IQCs, we are assigned to the appropriate peer groups and can check our bias in real time at any time using our software. We also agree with Reviewer #1 that the assessment of means and SDs for multiple instruments is a hot topic. We have expanded the discussion on this topic in section 16.

Comment 9: Line 212 - transient should be random

Response 9: We do not understand the objection. We are sorry. In section 7, we distinguish between “transient” and “permanent” out-of-control conditions on the one hand and “constant” and “random” error on the other hand (e.g., please be also referred to CLSI C24 or Badrick T and Loh TP, Clin Biochem 2023;114:39-42). We do not understand why we should replacetransient” by “random” in line in the second paragraph of section 7.

Comment 10: Table 1 - there are extra spaces in the Table between words.

Response 10: Reviewer #1 is right. When converting our Word document into the publisher's document, a format change unfortunately occurred. We have therefore now formatted Table 1 differently in the second version of our manuscript. We hope that Table 1 will now also be displayed correctly in the publisher's view.

Comment 11: There is no need to reproduce the Westgard rules in Figures - they are well known and can be referenced - suggest delete Figures 3,4, 5, 6, 7.

Response 11: We did as suggested and omitted Figures 3-7.

Comment 12: Line 314 - repeating the QC sample once, is the most powerful QC rule there is (Should I repeat my 1:2s QC  - Parvin et al)

Response 12: We know the mentioned paper by Parvin et al. Clin Chem 2012;58:925-929. Indeed, we cited this paper already in the first version of our manuscript (in another context). However, to our knowledge, the “1-2s repeat rule” is considered obsolete. Nowadays, using this rule as a rejection rule is not recommended any more (e.g., CLSI C24). We have therefore left the paragraph as it was in the revised version of the manuscript. We hope that Reviewer #1 sees it the same way we do.

Comment 13: Line 357 - I would not recommend using manufacturers target values. These need to be laboratory defined.

Response 13: We completely agree with Reviewer #1 on this point. Perhaps, the respective sentence in the first version of our manuscript was misunderstanding. We clarified this in the revised manuscript in section 9.

Comment 14: Line 394 - noncommutability of QC material is a problem - (Past, present, and future of laboratory quality control: patient- based real-time quality control or when getting more quality at less cost is not wishful thinking Katayev and Fleming).

Response 14: We fully considered this point raised by Reviewer #1 in our revised manuscript. We discussed this issue in the newly introduced section 16 of our revised manuscript.

Comment 15: There should be a section on the limitations of QC strategies - cost, false rejections (possible 80% of flags), where QC samples are placed, frequency, not responding to flags, not repeating patient samples - Howanitz et al Clinical laboratory quality control: A costly process now out of control; Westgard The 2017 great global QC survey.

Response 15: This criticism of Reviewer #1 is in line with that of Reviewer #2 (see below). In response to this comment, we discussed the issues raised by Reviewer #1 in the section 16 of our revised manuscript.

Comment 16: Suggest that the Abstract not be a repeat of sentences from within the text.

Response 16: We have left the abstract as it was. We have changed the text in the revised version of our manuscript according to this recommendation of Reviewer #1.

Reviewer 2 Report

Comments and Suggestions for Authors

The manuscript by L. Gruberand co-authors aims to summarize present data on internal quality controls in the medical laboratory. The authors thoroughly and widely discuss the topic with important examples and arguements. I find this article interesting and up to date.

I have just few recommendations on how this article may be improved:

The authors are expected to discuss the perspectives of the internal quality control development in medical health care system. It would be useful to highlight the weak points, problems and limitations of internal quality control in present time.

In conclusion, the article is well-written and may be accepted after minor corrections.

Author Response

Comment 1: The manuscript by L. Gruber and co-authors aims to summarize present data on internal quality controls in the medical laboratory. The authors thoroughly and widely discuss the topic with important examples and arguments. I find this article interesting and up to date.

Response 1: We would like to thank Reviewer #2 for this positive comment. No action is required for the revised version of our manuscript.

Comment 2: I have just few recommendations on how this article may be improved: The authors are expected to discuss the perspectives of the internal quality control development in medical health care system. It would be useful to highlight the weak points, problems and limitations of internal quality control in present time.

Response 2: We agree with Reviewer#1 that this could be an interesting extension of our work. Thus, we fully considered this point raised by Reviewer #2 in our revised manuscript. We discussed these issues in the section 16 of our revised manuscript.

Comment 3: In conclusion, the article is well-written and may be accepted after minor corrections.

Response 3: We also grateful for this positive statement from Reviewer #2. No action is required for the revised version of our manuscript.

Round 2

Reviewer 1 Report

Comments and Suggestions for Authors

The authors have revised the paper based on the Reviewers suggestions